# A Flexible Dynamic Reliability Simulation Approach for Predicting the Lifetime Consumption of Extravehicular Spacesuits during Uncertain Extravehicular Activities

**Yuehang Sun, Yun-Ze Li \***  **and Man Yuan**

School of Aeronautic Science and Engineering, Beihang University, 37 College Rd., Haidian District, Beijing 100191, China; sunyuehang@buaa.edu.cn (Y.S.); manyuan@buaa.edu.cn (M.Y.)
\* Correspondence: buaalyz@163.com; Tel.: +86-010-8233-8778

**Abstract:** The special use environment and uncertainty of extravehicular activities (EVAs) make it difficult to predict the lifetime consumption of extravehicular spacesuits in the traditional way. This paper presents a flexible reliability dynamic simulation model to predict the life loss of extravehicular spacesuits. Based on the images of traditional reliability change curves, new life assessment parameters, based on geometric analysis, are proposed as indicators of spacesuit life loss. Multiple influence factors are used to correct the spacesuit failure rate. The results of the study show that mission intensity is the main factor affecting the health status of the spacesuit, and the higher the mission intensity, the higher the failure rate. Additionally, the more frequently the spacesuit is used, the more times it is available, however, the overall service time will decrease. Concentrating on the mission at an early stage would lead to a significant and irreversible loss of life. Reliability is higher when more intense work is scheduled later in the EVA. Therefore, it is important to rationalize the mission duration, frequency, and work intensity of spacesuits. These reliability models predict the health status of the spacesuit and assist in optimizing the scheduling of EVA.

**Keywords:** extravehicular spacesuit; extravehicular activity; reliability analysis; lifetime consumption



## 1. Introduction

The extravehicular activity (EVA) schedule is planned in advance, but the exercise load during the activity is unpredictable. Therefore, the effect of EVAs on the lifetime consumption of extravehicular spacesuits has a large uncertainty and cannot be solved by analytical expressions [1]. Almost every part of the extravehicular spacesuit is of the highest technology and is costly to build. Once the astronauts have left the spacecraft, the only protection measure is the extravehicular spacesuit [2,3]. Spacesuits working in low-orbit environments are subject to degradation from dust, radiation, and other factors [4,5]. One of the major threats to future space exploration and utilization is the reduction of space security and sustainable use [6]. The proper functioning of the spacesuit is crucial for the safety of the astronauts. Therefore, it is crucial to apply flexible and effective methods for reliability assessment and life prediction of extravehicular spacesuits [7].

EVA is one of the most dangerous activities in human space exploration [8]. Astronauts need to acquire new information flexibly and adapt to changing environmental conditions [9]. As EVAs perform increasingly complex tasks, the demands on extravehicular spacesuits are becoming more demanding. This has led to continuous advances in extravehicular spacesuits and their systems [10]. The spacesuit will be used many times, and its lasting operation depends on regular testing and maintenance. Verify that the spacesuit is properly used and maintained in good operating condition during EVAs. After each EVA, the crew cleans the spacesuit of contaminated areas and stores them in a dry state [11]. To improve reliability and service life, the Orlan spacesuit uses replacement consumables, such as the introduction of removable filters in the sublimator [12]. Furthermore,

excessive replacement parts and frequency of replacement increase the preparation time before spacesuit outbound activities [13]. Another way to improve reliability is to configure redundancies concerning all critical systems and components of the spacesuit. The use of redundancy strategies is an effective measure to improve system reliability, however, there are constraints of cost, weight, and volume [14]; as such, consumables that can be regenerated become a better choice. Manned space vehicles must achieve high levels of performance within the strict constraints of cost, mass, power, and volume budgets [15]. Stringent safety and reliability requirements must be met to ensure mission and safety constraints. Reliability studies can provide a basis for spacesuit upgrades that balance safety, efficiency, and low consumption.

Extravehicular spacesuits are characterized by high cost, structural complexity, and long development times [15]. The long-term operation of the International Space Station confirms the importance of the proper use of spacesuits for EVAs [16]. During an EVA in 2013, the mission was prematurely terminated after a helmet water ingress malfunction occurred. The cause of the incident was water separator contamination. A similar incident of water ingress into the helmet occurred again in 2016 [17]. The normal operation of all functions of a spacesuit plays a vital role in the success and safety of the mission. Safety always comes first, so there is a strong need for risk management. Reliability, availability, maintainability, and safety are the key indicators for risk assessment. They, therefore, become important reference factors and objects of research [18]. There are several different approaches to reliability analysis, which mainly include building mathematical models, multiple regression tests, correlation coefficient analysis, and Bayesian statistics [19,20]. The accident model has been applied in military, aerospace, and transportation fields to better investigate, analyze, and prevent accidents [21]. Reliability prediction is introduced into the design process of aerospace missions for hazard analysis, critical component identification, and failure mode prediction [22]. Elisabeth et al. proposed a probabilistic risk analysis model for orbital particle impact on spacesuits using particle flux data [23].

According to current research on the reliability of interplanetary missions, the spacesuit life support system is responsible for around 38% of failures during EVAs. Then NASA presented a failure tree for the spacesuit life support system and listed the failure rate of the storage system as $0.0000372 \, \mathrm{d}^{-1}$ [24]. Some research summarized the abrasion of spacesuits, with boots and gloves being the most severe [25]. According to the International Space Station analysis, future astronauts will require around three hours per day to maintain the environmental control and life support systems [26]. The system performance, reliability, and maintainability should be weighed against the limitation of personnel ability and the technical level. The efficiency and safety of astronauts performing the intended operations are determined by the activity conditions, ergonomic characteristics, and reliability [27]. The number of times the spacesuit is available depends on factors such as the intensity and duration of the planned activity. The amount of maintenance and preparation activities before EVAs also varies depending on the state and interval of the EVA.

Emergencies in which the spacecraft environment is damaged require that the space-suit are able to take over all life support mechanisms. Predicting the probability of spacesuit failure is therefore critical to mission success and crew safety [28]. During EVAs, the effects of various factors on the spacesuits are always changing and accurate information cannot be obtained in advance. Therefore, the traditional reliability analysis methods cannot accurately predict the health status of spacesuits. In this paper, a new method based on a numerical integration solution is adopted. The spacesuit whole life cycle reliability model and the failure rate correction model are developed. The effects of mission frequency and intensity on spacesuit reliability are investigated to provide an optimized solution for EVAs.

## 2. Conception and Technical Approach

### 2.1. The Necessity and Difficulties of Spacesuit Life Prediction

The life span of an extravehicular spacesuit is unpredictable in an EVA and full of uncertainties. This is because it is influenced by a combination of factors including

pressurization, wear, activity, and temperature. These factors affect the life span of the spacesuit at both the time and frequency levels, as shown in Figure 1. The influencing factors were divided into two categories and studied in different ways. Extravehicular spacesuits are very delicate and complex systems, and the reliability analysis for spacesuit life prediction is not only important but also challenging.

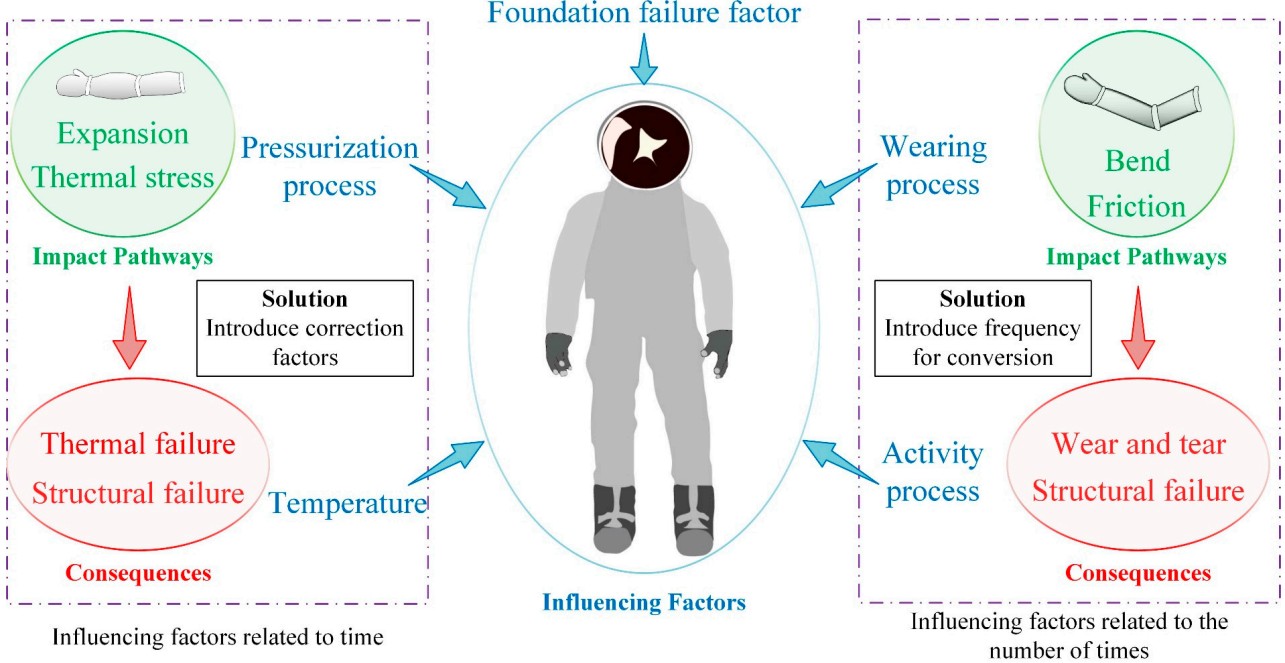

**Figure 1.** Factors and ways that influence the reliability of spacesuits.

The number of pressurizations, wears, and joint movements affect the health status of the spacesuit at the frequency level. Extravehicular spacesuits are stored inside the cabin and are unpressurized. The spacesuit is pressurized before the mission and then worn by the astronauts. With increased use, the spacesuit wears out. Spacesuits use multiple layers of material; the inner layer that maintains the pressure is a polyurethane-coated nylon pressure suit, the outer layer includes neoprene-coated nylon lining, aluminized polyester film insulation, and the outer fabric. In order to give astronauts good mobility, the spacesuit usually uses soft joints in the shoulders, elbows, wrists, and knees. Soft joints have a high tensile strength and are made of fabric, rubber, and polymer. Extravehicular missions require the astronaut to perform delicate activities, so there is a lot of movement in the joints, as such, the design of the soft structure part is very important [29,30]. The joints are too soft for protection and too hard for flexibility. The soft structures are subject to severe wear and degradation because the joints rub most frequently during the crew's movements. In addition, the failure of the airtight layer of the joints is mainly in the form of bending and crushing.

Temperature affects the health status of the spacesuit at the temporal level. Spacesuits contain a large number of materials that are significantly affected by temperature, including polyurethane coatings, neoprene coatings, and polyester films, so the temperature is also an important factor affecting the life expectancy of a spacesuit. Regarding the study of product degradation, the Arrhenius equation is the most commonly used acceleration model with a wide range of applications [31]. According to the Arrhenius equation, the higher the temperature, the faster the reaction rate within a certain range. High temperatures can cause thermal aging of the material and a reduction in reliability. Spacesuits are normally stored at the right temperature. However, both the external and internal environments can influence the temperature of the spacesuit during use, causing additional thermal failure.

## 2.2. Introduction of Spacesuit Structure System

The extravehicular spacesuit is a personal protective, life-saving device that protects the lives of astronauts during space missions. The spacesuit consists of a pressure garment, an aerospace helmet, aerospace boots, and pressure gloves. The specific structure and function of the spacesuit are shown in Figure 2. The pressure garment is the main body of the spacesuit, which must be well sealed and allow for a certain range of movement, with a high level of protection and performance. Astronauts are required to wear spacesuits for space missions, so as a miniature spacecraft, the joints of the extravehicular spacesuits are mostly made of flexible soft structures. Joint torque is critical to crew comfort, fatigue, productivity, and medical impact [32].

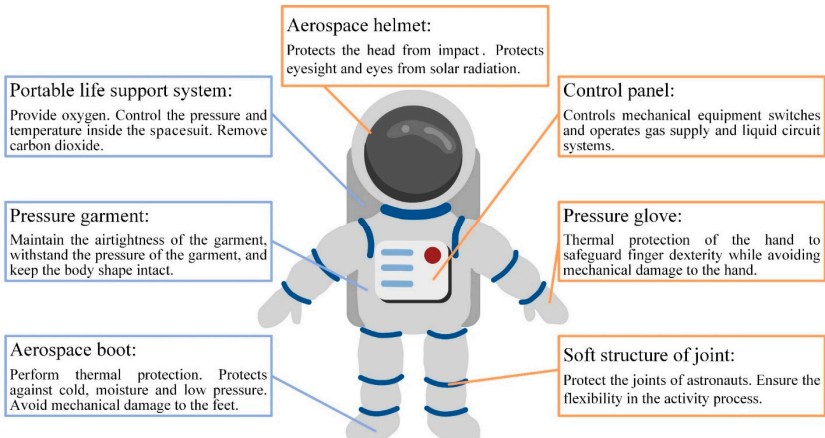

**Figure 2.** Spacesuit system construction components.

The main functions of the spacesuit include keeping the astronaut's body temperature and pressure stable, blocking harmful radiation, providing oxygen, and removing carbon dioxide. Humans will die if they leave the spacesuit and are exposed to the space environment for 30 s. The proper use of the extravehicular spacesuit is therefore particularly important in the complex and harsh environment of space. Spacesuits are very complex and difficult to develop. The cost of a spacesuit is very high. In some countries, parts that are prone to wear and tear are replaced for use to extend the life of spacesuits. It is therefore important to assess the health of the spacesuit in time, to predict the remaining service life, and to determine whether it is ready for further use. It helps to rationalize the intensity of the mission and to provide timely care and maintenance of components.

## 2.3. Foundation Degradation Failure of Spacesuit

Indicators commonly used in traditional reliability analysis include failure rate, reliability, and average life. The failure rate is used to describe the degree to which the product fails and is one of the common quantitative characteristics of product reliability. Degradation failure is the gradual and slow decline in the functional characteristics of a product over time during storage or operation until it fails to function properly. Basic failure is a continuous process and is usually monotonic. Once the spacesuit has degraded to a failure state, it is no longer able to protect and sustain the astronauts in their work.

Reliability is an important quality characteristic of a product and plays an extremely important role in meeting the needs of the user. The higher the reliability of a product the longer it can be considered to work. The object of traditional reliability analysis methods is the time to failure of a product. However, the probability of failure in a short period of time is small for a highly reliable and costly product such as the spacesuit, so the failure data is limited. To solve this problem this paper proposes new life assessment parameters based on reliability images.

The unusable life loss (ULL), the life of points reached (LPR), and the predicted usable life (PUL) are presented as life parameters of the spacesuit and are represented in Figure 3. Area 1 is the ULL, which represents the life consumption at the current moment due to some damage. Area 2 is the LPR, indicating the service life of the spacesuit at the current moment. Area 3 is the PUL, which predicts the remaining service life of the spacesuit. These three life parameters are derived from the reliability image and can be used as important indicators of reliability.

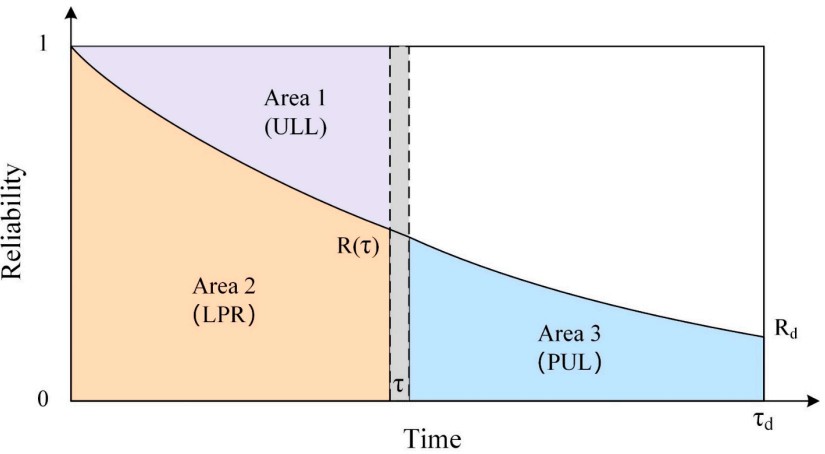

**Figure 3.** Life parameter information in reliability curve.

### 2.4. External Factors Affecting EVA Spacesuit Lifetime Consumption

The factors of friction, bending, and heat during the mission of a spacesuit all have an impact on the foundation failure rate. The factors that correct for the failure rate of the spacesuit during a mission are divided into four categories: number of pressures, number of wears, number of joint movements, and temperature. The specific failure rate can be obtained by adding the four correction factors to the foundation failure rate of the spacesuit, as shown in Figure 4. The correction of the failure rate not only gives the lifetime of the spacesuit in long-term storage but also reflects the effect of repeated use on the lifetime, making the study of the reliability of the spacesuit in the whole life cycle and extravehicular stage more relevant to the real situation.

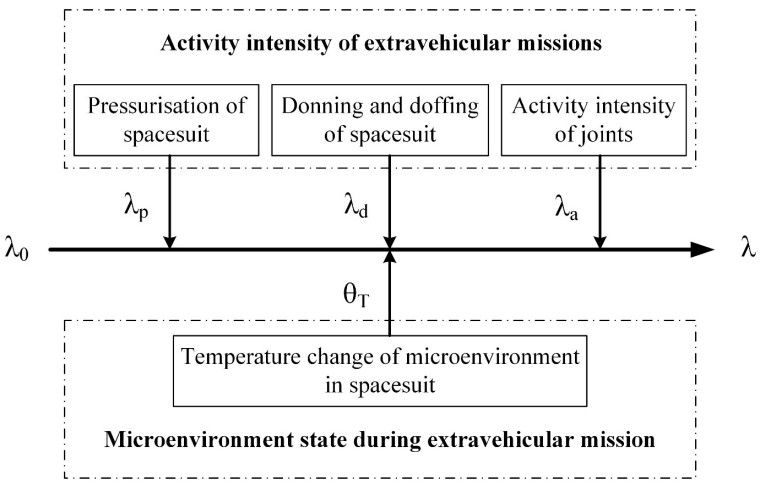

**Figure 4.** Spacesuit failure rate correction process.

## 3. Dynamic Model

Some assumptions are made about the study before modeling. 1. Temperature and the number of wears, pressurization, and activities are the main influencing factors of the spacesuit failure rate; 2. The activity intensity remains consistent over a short period of time; 3. The basic failure rate of the spacesuit obeys the Weibull distribution; 4. The temperature inside the spacesuit is uniform.

### 3.1. Modified Model of Reliability Indicators for Spacesuit Structure

The correction of the foundation failure rate of the spacesuit provides the basis for the subsequent solution of other reliability indicators. The spacesuit, which is always in storage, should be pressurized, worn, and bent when in use. Three additional failure rates need to be added to the foundation failure rate $\lambda_0(\tau)$. The correction for temperature utilizes a temperature correction factor to reflect the effect of temperature on the reliability indicators. The temperature correction factor $\theta_T$ is multiplied by the sum of the component failure rates to obtain the specific failure rate of the spacesuit during the mission. Where the wear and bending of the spacesuit are related to the number of times. To investigate the effect of both on the failure rate in time, the frequency was introduced. The failures related to the number of times were converted to the time dimension. In addition, the risk of failure of spacesuits increases by a factor of about two under pressurized operation compared to static storage [33].

$$\lambda(\tau) = \left( \lambda_0(\tau) + \lambda_p(\tau) + \lambda_d(\tau) + \lambda_a(\tau) \right) \times \theta_T \tag{1}$$

$$\lambda_d(\tau) = \lambda_d(n) \times f_d \tag{2}$$

$$\lambda_a(\tau) = \lambda_a(n) \times f_a \tag{3}$$

where $\lambda(\tau)$ is the specific failure rate; $\lambda_p(\tau)$ is the pressurization failure rate; $\lambda_d(\tau)$ is the wearing failure rate; $\lambda_a(\tau)$ is the activity failure rate; $\lambda_d(n)$ and $\lambda_a(n)$ are the failure rate for the number of wears and activities, respectively; $f_d$ is the wearing frequency of the spacesuit; and $f_a$ is the frequency of joint movement.

Both the airtight limiting structure and the thermal barrier coating of spacesuits are subject to temperature degradation. Moreover, a large number of rubber structures are used in spacesuits, such as polyurethane and neoprene coatings. The aging of most rubber materials can be described by the Arrhenius equation, and therefore the Arrhenius equation applies to the rate of thermal aging of most spacesuit materials as a function of temperature. The temperature correction factor is solved according to the Arrhenius equation, which reflects the relationship between the chemical reaction rate constant and temperature [34,35].

$$\theta_T = \exp\left( -\frac{A}{T} + B \right) \tag{4}$$

$$A = \frac{Ea}{k} \tag{5}$$

where Ea is the activation energy, k is Boltzmann's constant, T is the average temperature of the spacesuit, and B is a given characteristic constant.

### 3.2. Basic Reliability Indicators Model

Products will use scientific indicators to measure product reliability. Commonly used reliability indicators include reliability, failure rate, average operating time, etc. Reliability R(t) is defined as the probability that a product will complete its function under specified

conditions and within a specified time. The relationship between the reliability function and the failure rate is shown below:

$$\lambda(\tau) = -\frac{\mathrm{d}\ln R(\tau)}{\mathrm{d}\tau} \tag{6}$$

Failure rate and reliability as health status assessment values indicate the health of the extravehicular spacesuit. The mathematical models reflecting the failure law will vary depending on the failure mechanism and failure mode of the product. Different probability distributions need to be used for different objects to fully reflect reality. The Weibull distribution is a relatively complex distribution widely used in reliability analysis. It is a good fit for all types of test data and is widely used in the study of electronic product reliability. In this paper, the Weibull distribution is used as the distribution of the failure rate of spacesuits, and the model is as follows:

$$\lambda(\tau) = \frac{\omega\tau^{\omega-1}}{\alpha^{\omega}} \tag{7}$$

where $\omega$ is the shape parameter and $\alpha$ is the scale parameter.

### 3.3. Life Parameter Model of Spacesuit

Life parameters are also important indicators of reliability and include ULL, LPR, and PUL. The solution for the life parameters can be calculated using geometric methods based on reliability curves, respectively. ULL is the area above the reliability curve, which can be obtained by integration. The sum of ULL and LPR is the current moment. The specific mathematical model is therefore as follows:

$$\frac{\mathrm{dULL}}{\mathrm{d}\tau} = 1 - R(\tau) \tag{8}$$

$$\mathrm{LPR} = \tau - \mathrm{ULL} \tag{9}$$

PUL is a prediction of the remaining usable life for a known design life and is an approximate prediction parameter. We propose two different solving methods to predict PUL, the first one is by calculating the approximate area of the trapezoid.

$$\mathrm{PUL} = 0.5 \times (R(\tau) + R_{\mathrm{d}}) \times (\tau_{\mathrm{d}} - \tau) \tag{10}$$

where $\tau_{\mathrm{d}}$ is the design life and $R_{\mathrm{d}}$ is the reliability corresponding to the design life.

The second is the constant reliability rate of the change estimation method. The estimation is carried out assuming a constant rate of change in reliability after the current moment, setting $\gamma(\tau)$ as the rate of change in reliability. Assuming that $\gamma(\tau)$ is constant, the time to reach the terminal reliability can be calculated. Finally, the PUL is solved according to the geometric method. The equation is as follows:

$$\gamma(\tau) = \frac{\mathrm{d}R(\tau)}{\mathrm{d}\tau} = \frac{\mathrm{d}R(\tau)}{\mathrm{d}x} \times \frac{\mathrm{d}x}{\mathrm{d}\tau} = -R(\tau) \times \lambda(\tau) \tag{11}$$

$$x = -\int_0^\tau \lambda(\tau)\mathrm{d}\tau \tag{12}$$

$$\tau_{\mathrm{p}} = \frac{R(\tau) - R_{\mathrm{d}}}{|\gamma(\tau)|} = \frac{R(\tau) - R_{\mathrm{d}}}{R(\tau) \times \lambda(\tau)} \tag{13}$$

$$\mathrm{PUL} = 0.5 \times \tau_{\mathrm{p}} \times (R(\tau) + R_{\mathrm{d}}) \tag{14}$$

where $\tau_p$ is the time to reach the terminal reliability. In order to make a conservative estimation, the second constant reliability rate of change method is used for estimation in this paper.

### 3.4. Simulation and Characteristic Parameters Setting

The simulation program is written and calculated in Visual Studio 2019. The above mathematical model requires several characteristic parameters to be given. The parameter settings for the simulation of the spacesuit reliability indicators are shown in Table 1.

**Table 1.** Parameter setting for spacesuit reliability simulation.

| Parameters | Units | Values | Parameters | Units | Values |
|---|---|---|---|---|---|
| $\alpha$ | / | $2.1 \times 10^8$ | $\omega$ | / | 1.36 |
| $\lambda_d(n)$ | / | $1/40$ | $\lambda_a(n)$ | / | 1/40,400 |
| $R_d$ | / | 0.125 | $\tau_d$ | y | 5 |
| k | ev·K$^{-1}$ | $8.62 \times 10^{-5}$ | Ea | ev | 1 |
| B | / | 38.76 | - | - | - |

The parameter settings are based on the new generation of Feitian extravehicular spacesuits developed in China, which are designed to be used 15 times in 3 years [36]. The design margin is considered in the simulation. The design life of the spacesuit is set to 5 years, the number of wearable times is 40, and the maximum number of joint movements is 40,400, beyond which it is considered unusable. This is used to determine the frequency-based failure rate in the parameters. In addition, the activation energy is the energy required to overcome to start a certain physicochemical process, setting the activation energy for the thermal failure of the spacesuit to 1ev.

## 4. Results and Discussion

### 4.1. Lifetime Consumption of the Spacesuit in Long-Cycle Situation

The proper scheduling of tasks has an important impact on extending the life of a spacesuit. Apart from consumables and customized components, the expensive extravehicular spacesuits can be used many times over. Spacesuits that are not in use are stored. In order to study the reliability performance of the spacesuits over the designed service life, the full life-cycle reliability indicators are simulated. The health of the spacesuit at each moment is simulated to assess and optimize the mission schedule. This section examines the reliability of spacesuits under different mission arrangements:

#### 4.1.1. Uniform Arrangement of Extravehicular Tasks

Simulate the change in reliability indicators of the spacesuit over a three-year period. Schedule four missions per year with the same interval between each mission. Reliability indicators were also compared for the four operating conditions: storage state, light task intensity, medium task intensity, and heavy task intensity.

As shown in Figure 5, the failure rate of the spacesuits that have been in storage is much lower than in the other three cases, with a reliability of around 0.7 after three years. There is a sudden increase in the spacesuit failure rate during each extravehicular mission due to other failure factors that are added during the mission. The overall trend in failure rates for the three different activity intensities is similar to the Weibull distribution in the storage state, with a faster increase in failure rates in the early stages. The extravehicular failure rate for the corresponding number of times per year is slightly lower in the second year than in the first and third years. The higher the mission intensity the higher the failure rate and the faster the reliability decreases. High-intensity work increases the number of activities in the joint areas, increasing the risk of wear and tear, which affects the life of the spacesuit. It is therefore important to control the intensity of each EVA reasonably.

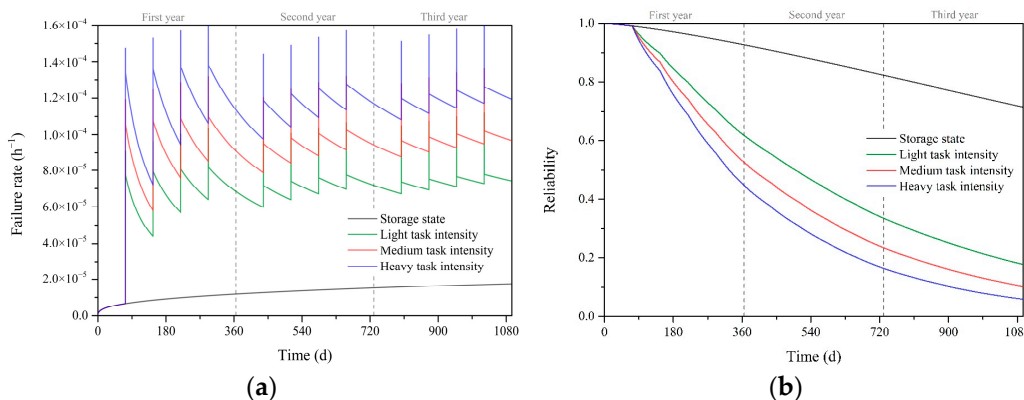

**Figure 5.** Spacesuit reliability indicators for a uniform mission schedule. (**a**) Failure rate. (**b**) Reliability.

As shown in Figure 6, the spacesuit has the smallest ULL in storage, indicating the least amount of usable life lost in three years. The loss of life is a continuous state. As the number of uses increases and the intensity of the activity increases, more and more life is lost and the ULL increases with time. The faster the life is lost, the faster the ULL increases due to the higher rate of failure in the later period, eventually reaching a relatively constant rate of increase. The LPR is the opposite of the change in ULL. When the failure rate is small, more life can be used effectively and the LPR is larger. Thus the spacesuit can be used more efficiently under lighter missions. Mission scheduling requires a balance between efficiency and safety. As PUL is strongly related to the failure rate, PUL also reacts to sudden changes in the failure rate each time. Scheduling high-intensity tasks all the time will accelerate life wear with a small PUL.

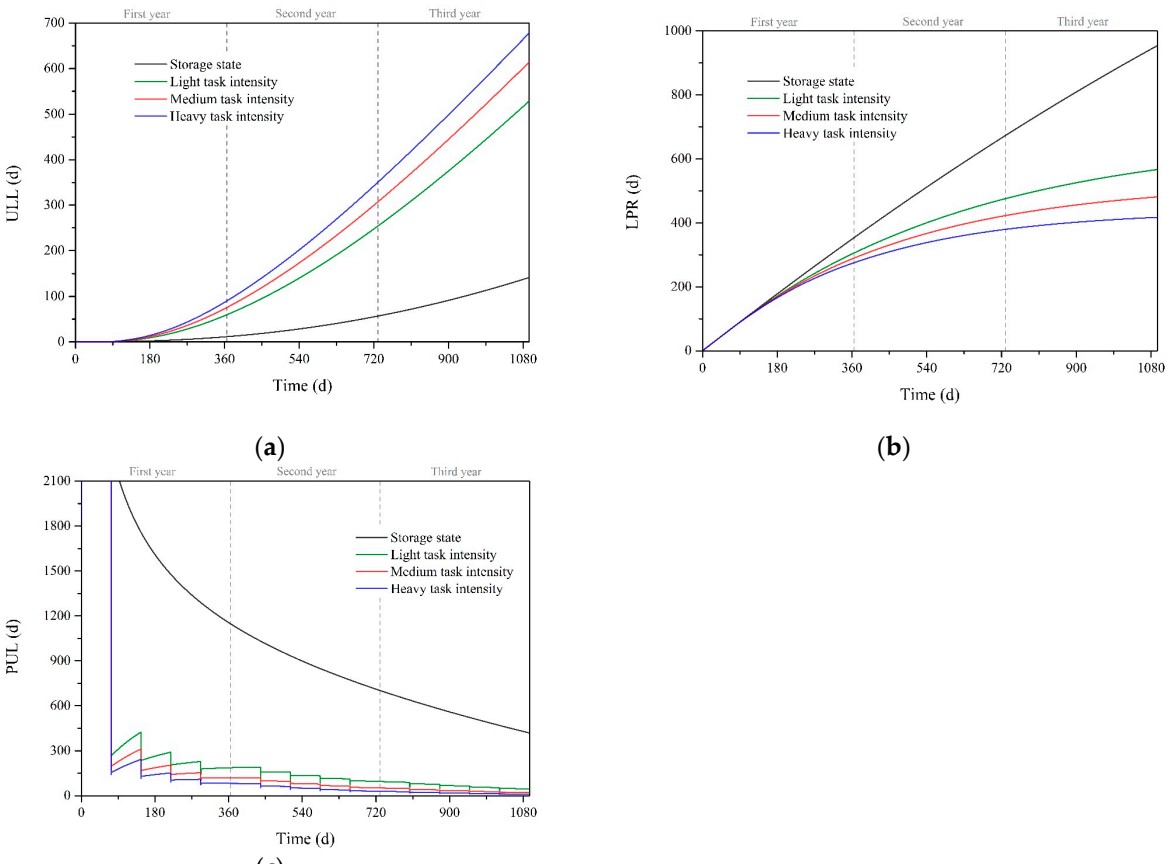

**Figure 6.** Spacesuit life parameters for a uniform mission schedule. (**a**) ULL. (**b**) LPR. (**c**) PUL.

### 4.1.2. Centralized Arrangement of Extravehicular Tasks

The tasks are then scheduled centrally. Twelve missions were scheduled in the first, second, and third years respectively, each at the same interval and with the same mission intensity. Compare the reliability indicators of the spacesuits under frequent use over a three-year period.

As shown in Figure 7, the failure rate is the highest when the tasks are concentrated in the first year, and the reliability rapidly drops to a low level in the early stage. This is mainly influenced by the frequency of wear and joint activity of the spacesuit. The concentration of extravehicular missions in the early years means that frequent wear and activity in a short period increase the risk of failure and therefore the failure rate is significantly higher than in the other two cases. In contrast, task scheduling is concentrated in the later stages, where reliability is higher. Although the foundation failure rate is higher in the later stages, the superimposed frequency-related failure rate is smaller. If the task is scheduled centrally in the third year, the reliability at the end of the task is around 0.5. Scheduling missions too centrally is detrimental to astronaut operations and increases the probability of unexpected events. Therefore missions have to be scheduled more rationally within the specified time.

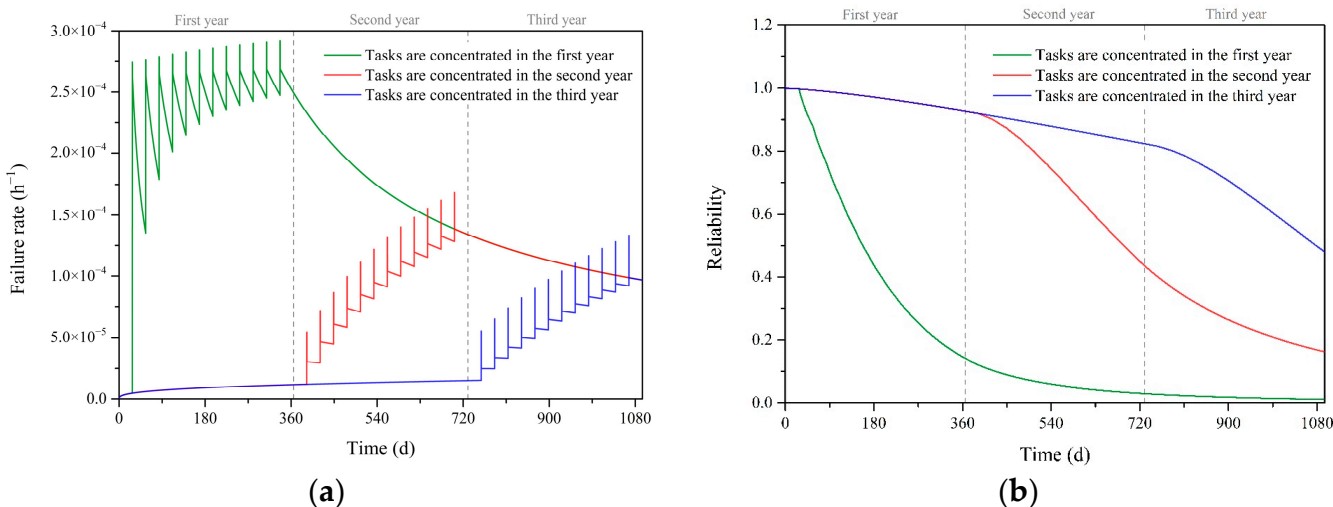

**Figure 7.** Spacesuit reliability indicators for a centralized mission schedule. (**a**) Failure rate. (**b**) Reliability.

Under the working condition of concentrated tasks in the first year, ULL has been increasing obviously, as shown in Figure 8. Less of the design life of the spacesuit is effectively utilized and the LPR is maintained at a small value. The PUL also drops rapidly in the early stages and is maintained at a low level. This is mainly because these life parameters are affected by high failure rates and low reliability, resulting in many irreversible losses and reduced life levels. With tasks all centrally scheduled in the second year, the ULL is much smaller than the first working condition and slightly larger than the third working condition. The advantage of task scheduling in the later stages diminishes as time increases. The LPR is also at a higher level in the second case, while the reliability at the end of the task is similar to that of the uniformly distributed light task case. A concentrated task schedule in the middle or later part of the design life has relatively good life indicators. Therefore try to avoid scheduling too many extravehicular missions in the early stages. The scheduling of extravehicular missions in different phases has a clear impact on the rational use of spacesuits.

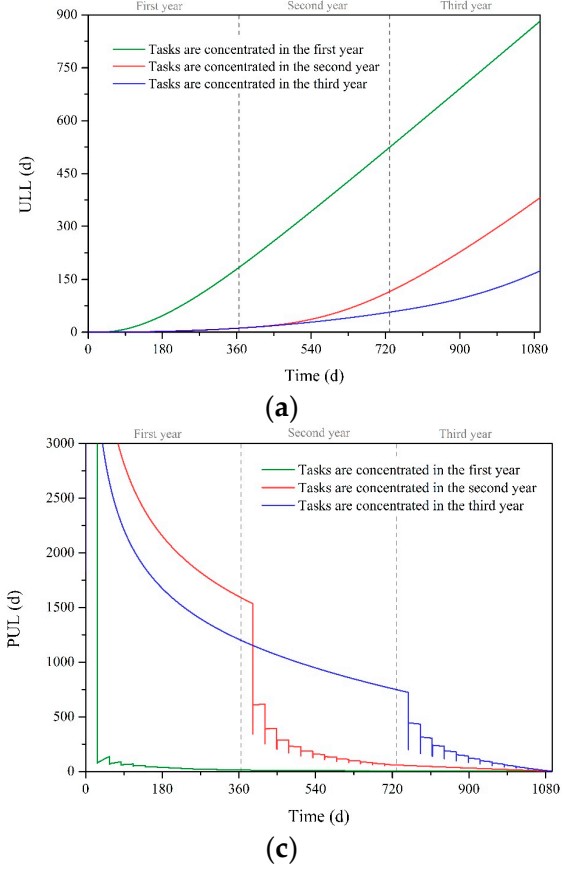
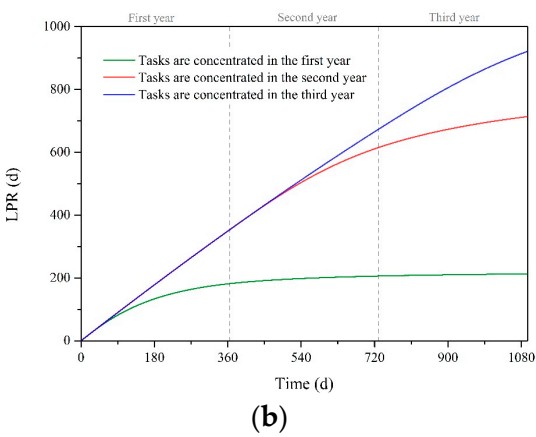

**Figure 8.** Spacesuit life parameters for a centralized mission schedule. (**a**) ULL. (**b**) LPR. (**c**) PUL.

### 4.1.3. Effect of Mission Frequency on Availability of Spacesuits

In the previous studies, the time interval between every two extravehicular missions was the same and the number of missions over the design life was kept consistent. This section examines the effect of the frequency of EVAs on the reliability indicators, with the same intensity of activities set each time. Three operating conditions, six times a year, four times a year, and twice a year, were selected. The question of the number of EVAs that can be performed under conditions of fixed termination reliability is investigated.

Set 0.3 as the termination reliability. If the reliability is less than 0.3, then the spacesuit is deemed unusable again. In the case of six EVAs in a year, the spacesuit is considered inoperable after the eighth mission when the reliability is less than the termination value. In the case of four EVAs a year, the spacesuit can be used for a maximum of seven missions. In the case of two EVAs a year, the spacesuit will become invalid after six missions. As shown in Figure 9d, as the frequency of EVA increases, the number of missions the spacesuit can perform increases. This is due to the risk of failure of the spacesuit during storage as well. At lower frequencies of EVA, the storage time of the spacesuit increases, the accumulated risk of failure increases, and more of the life is lost in the storage phase. Although performing EVAs at a low frequency reduces the number of times the spacesuit is used, the invalidation time of the spacesuit is delayed. With two EVAs a year, the spacesuit fails in about three years. With six EVAs a year, however, the suit has already been disabled for more than one year. The number of times the suit is used and the duration of use can therefore be balanced by adjusting the frequency of extravehicular missions. Choosing the right mission frequency also has a clear impact on the reliability profile of the spacesuit.

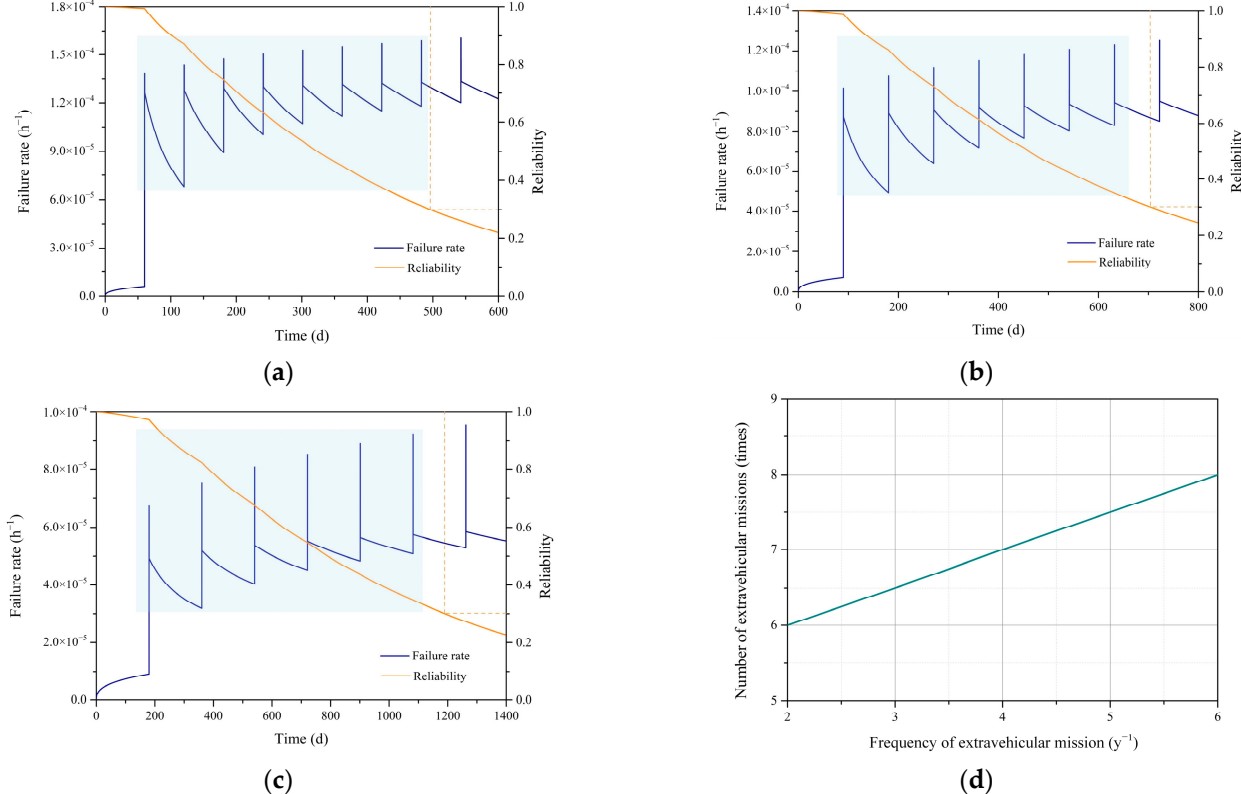

**Figure 9.** Impact of mission frequency on spacesuit reliability indicators. (**a**) Six extravehicular missions a year. (**b**) Four extravehicular missions a year. (**c**) Two extravehicular missions a year. (**d**) Available times of spacesuit.

*4.2. Reliability Indicators of Spacesuit under the Single Extravehicular Mission*

The full life-cycle reliability indicators of spacesuits have been analyzed. In addition to the effects of the long life cycle, the state of the spacesuits in the short term is also of interest. The following study examines the changes in the reliability status of the spacesuits during EVA. The first EVA was selected for the study, with a mission duration of eight hours. Based on the previous reliability study, the short-term simulation adds the influence of temperature correction and pressurization magnitude. The impact of the single mission schedule on the use of the spacesuit is further demonstrated in more detail.

4.2.1. Effect of Mission Intensity on Spacesuit Reliability Indicators

An EVA is divided into three phases: pre, mid, and post. Each phase is arranged with tasks of different intensities. Two working conditions were chosen for the comparison analysis: a gradual increase and a gradual decrease in the intensity of the task, as shown in Figure 10. The average temperature of the spacesuit is also slightly affected by the intensity of the mission. The increased heat production of the astronauts during heavy workloads can cause fluctuations in the temperature of the spacesuit. A temperature correction factor is used to reflect the effect of temperature fluctuations on reliability. At the same time, the spacesuit is re-pressurized before exiting the capsule during the actual mission. The spacesuit is first flushed with a high flow of gas during pressurization and the pressure is maintained at around 40 kpa during the subsequent EVA. The pressure inside the spacesuit is high for a short period of time before leaving the cabin when the effect of the pressurization failure rate is greater. The variation curves of some factors were assumed as shown in the picture. The above factors are added to the correction of the spacesuit reliability indicators and simulated.



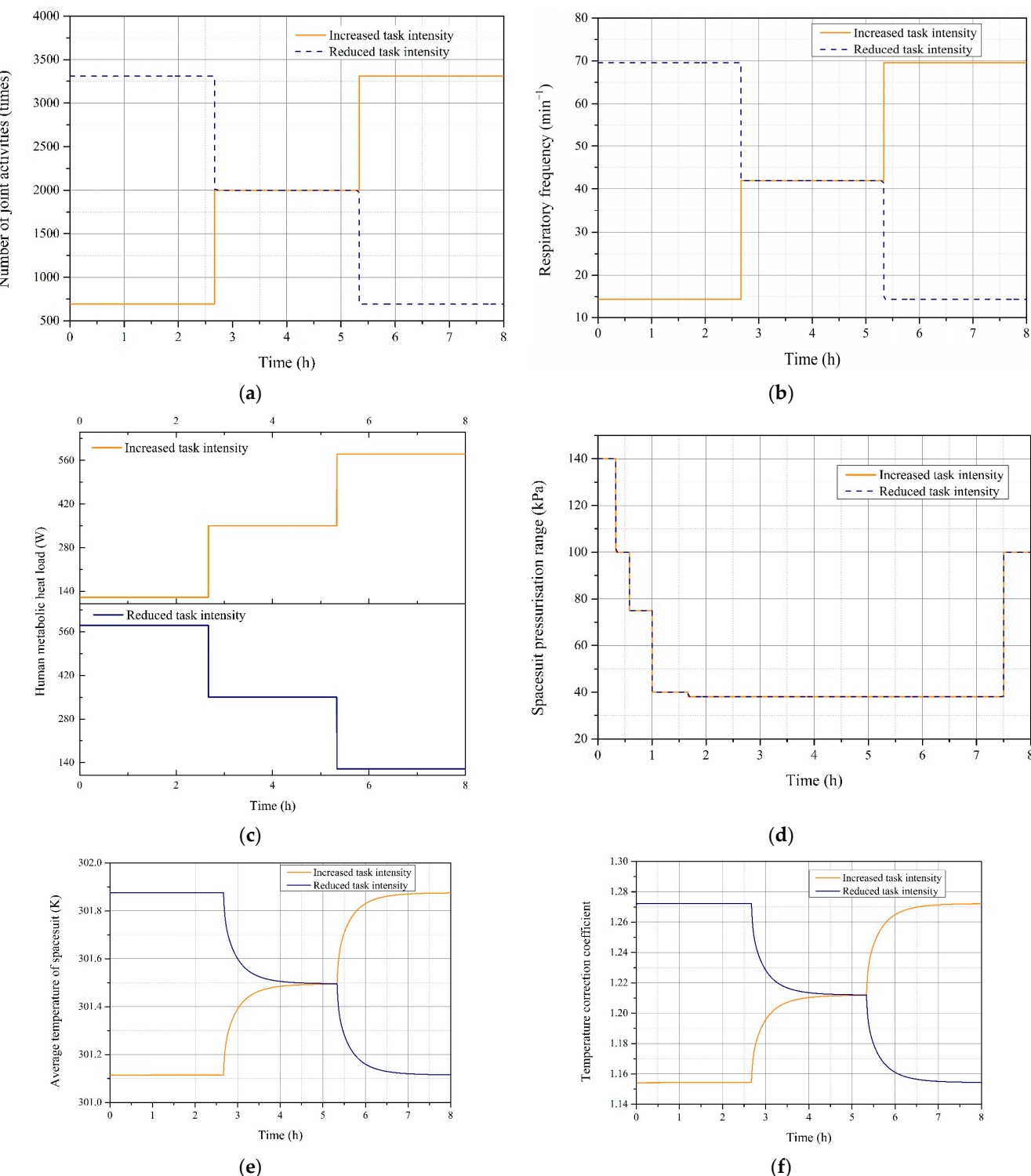

**Figure 10.** Failure rate correction factors of a single mission. (**a**) The number of joint activities. (**b**) Respiratory frequency. (**c**) Human metabolic heat load. (**d**) Spacesuit pressurization range. (**e**) The average temperature of the spacesuit. (**f**)Temperature correction coefficient.

As shown in Figure 11, during the first hour of EVA, the failure rate of the spacesuit is mainly influenced by the magnitude of pressurization. During subsequent EVA, the failure rate is mainly influenced by the intensity of work and temperature. As the failure rate increases, the rate of decrease in reliability also increases. At the end of the first extravehicular mission, the failure rate is somewhat greater for the first operating condition.

However, the terminal reliability of the second working condition is lower. Due to the heavy work carried out in the first period, the failure rate remains at a higher level. There is thermal inertia in the change in temperature and the temperature correction factor is consistently greater for the decreasing intensity condition in the early and mid-mission periods. So with these two factors, the failure rate of the increasing intensity condition exceeds that of decreasing intensity condition in the late stage of the task. The average failure rate is therefore greater for the second operating condition, resulting in a greater reduction in reliability. The difference in reliability between the two operating conditions is approximately 0.0002. This suggests that in a single EVA, scheduling the more intense work as late in the mission as possible is beneficial in maintaining higher reliability and extending the life of the spacesuit.

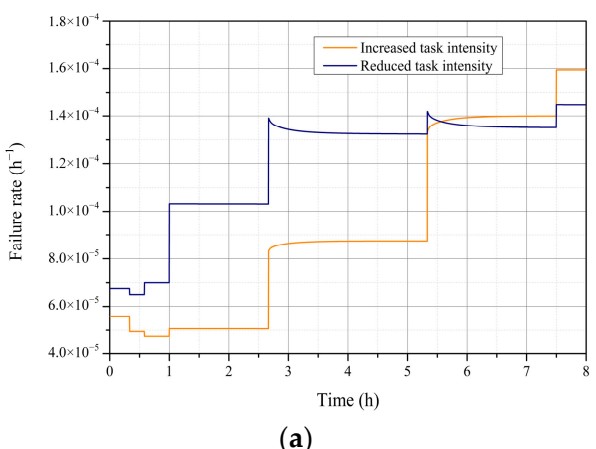

(**a**)

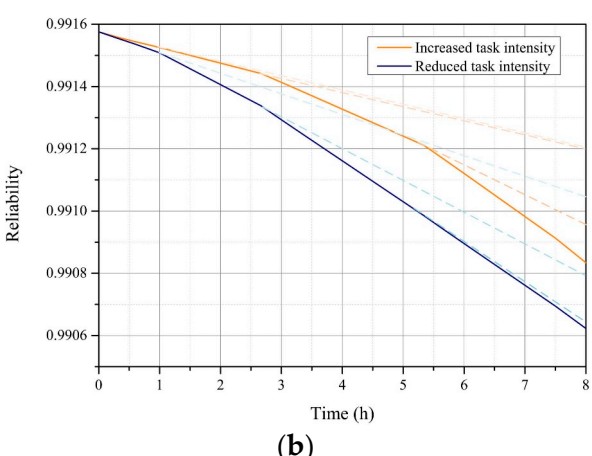

(**b**)

**Figure 11.** Spacesuit reliability indicators for different mission arrangements. (**a**) Failure rate. (**b**) Reliability.

4.2.2. Simulation of Single Task with Reference to Actual Working Conditions

Concludes the study of regular changes in task intensity. In practice, the activity intensity varies randomly. A design condition was selected to simulate the task intensity during the entire EVA. The variation of reliability indicators during the actual EVA was simulated. Less activity was performed during the pre-mission pressurization, followed by an increase in activity intensity during exiting the capsule. The activity intensity also increases during the final return phase of the mission. The intensity of the mission is represented using human metabolic heat, as shown in Figure 12a.

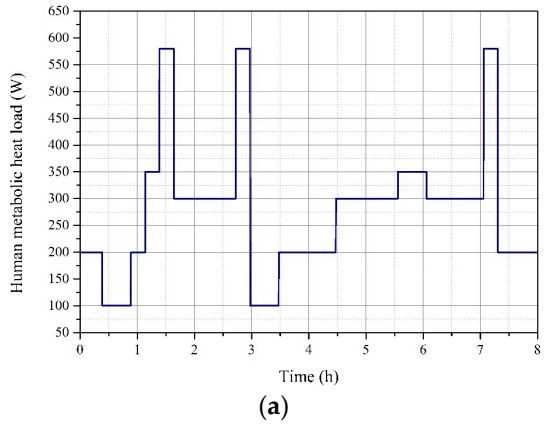

(**a**)

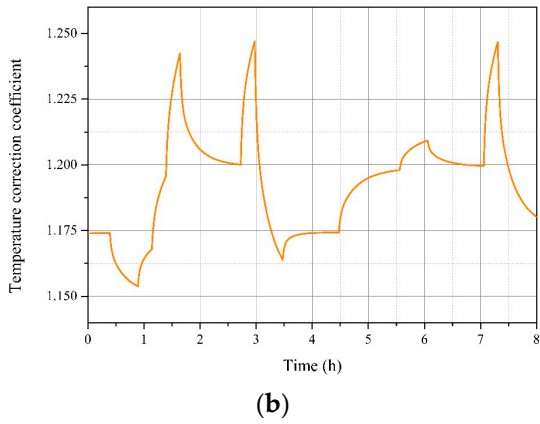

(**b**)

**Figure 12.** Correction factors for failure rate at design working condition. (**a**) Human metabolic heat load. (**b**) Temperature correction coefficient.

The failure rate of spacesuits shows an overall increasing trend over time, as shown in Figure 13. The increasing value of the failure rate is mainly influenced by the intensity of the activity. The incremental increase in the failure rate of the spacesuit is greatest when the body's metabolic heat is located at its maximum value. The average temperature mainly influences the trend of the failure rate per segment. The reliability of the spacesuit decreases with time and the rate of decrease increases slightly with time. The difference in reliability for this EVA was approximately 0.001.

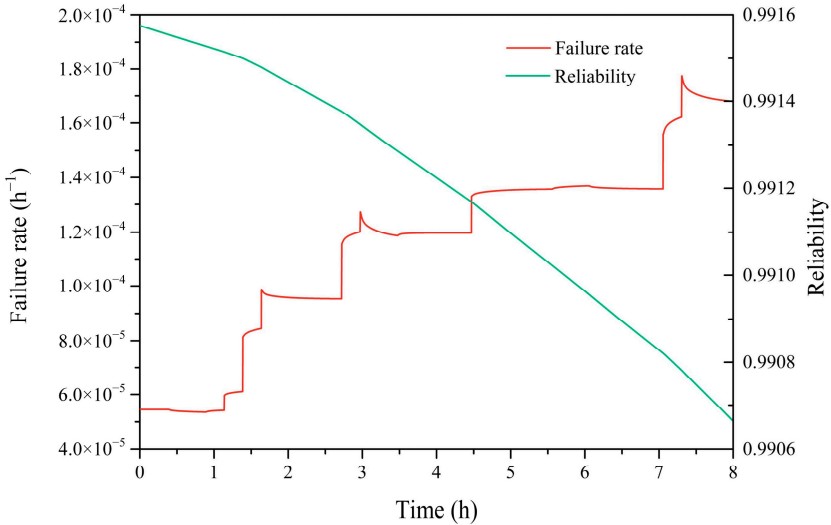

**Figure 13.** Reliability indicators for spacesuits under design conditions.

## 5. Conclusions

Keeping extravehicular spacesuits in good working order is essential for the safety of astronauts. In this paper, a simple and effective new extravehicular spacesuit reliability model is proposed. The changes in the reliability of the spacesuit throughout its life cycle and during a single EVA under different mission frequencies and intensities are investigated. The following conclusions are specifically obtained:

1.  The reliability of spacesuits decreases as the intensity of EVA increases. The higher the intensity of EVA, the more ULL and the less PUL.
2.  In the full life cycle study of spacesuits, concentrating EVAs into the early stages can result in significant lifetime loss. Scheduling too many extravehicular missions in the early stages should be avoided.
3.  As the frequency of EVA increases, the number of times the spacesuit is available increases, but the overall time available decreases.
4.  The failure rate of a spacesuit is mainly influenced by the intensity of the activity. The later the high-intensity work is performed in a single EVA, the more reliable the spacesuit becomes.
5.  The selection of the frequency of EVAs requires a balance between the number of times the spacesuit is used and the time requirements. A reasonable arrangement of the intensity, frequency, and duration of EVAs can improve the reliability of the spacesuit.

The intensity, duration, and frequency of missions during EVAs have a significant impact on the reliability of spacesuits. Reliability models allow for a more rational organization of tasks and improve the safety of spacesuit use without compromising efficiency. The spacesuit has a complex structure, and some random events may occur during the mission that cannot be reflected in the model, and further research will be conducted in the future to solve this deficiency.

**Author Contributions:** Conceptualization, Y.-Z.L.; Methodology, Y.-Z.L.; Software, Y.S.; Validation, M.Y.; Formal Analysis, Y.S. and M.Y.; Investigation, Y.S. and M.Y.; Resources, Y.S. and Y.-Z.L.; Data Curation, Y.S.; Writing—Original Draft Preparation, Y.S. and Y.-Z.L.; Writing—Review & Editing, Y.S. and M.Y.; Visualization, Y.S.; Supervision, M.Y. All authors have read and agreed to the published version of the manuscript.

**Funding:** This research received no external funding.

**Institutional Review Board Statement:** Not applicable.

**Informed Consent Statement:** Not applicable.

**Data Availability Statement:** All data and models used in this study are in the published article.

**Acknowledgments:** The authors are deeply grateful to the support of Astronaut Center of China.

**Conflicts of Interest:** The authors declare no conflict of interest.

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
