# Peer review of "A Flexible Dynamic Reliability Simulation Approach for Predicting the Lifetime Consumption of Extravehicular Spacesuits during Uncertain Extravehicular Activities"

_aerospace, doi:10.3390/aerospace10050485_

Round 1

Reviewer 1 Report

·         The abstract could benefit from a more detailed explanation of the methodology used in the research. For example, it would be helpful to know more about the specific graphs used to evaluate the remaining service life of spacesuits.

·         The abstract could be improved by including more information about the results of the study. While the abstract mentions that the results show a higher level of reliability when more intense work is scheduled later in the extravehicular activity, it would be useful to know more about the magnitude of this effect and how it was quantified.

·         It would be helpful to know more about the limitations of the study and any assumptions that were made in developing the simulation model. This information would help readers to better understand the scope and applicability of the research.

·         Use subheadings to make the section more organized and reader friendly.

·         The Introduction section could benefit from more concise and clear writing. Some sentences are quite long and complex, making it difficult for the reader to follow the main points.

·         It would be helpful to provide more context for some of the references cited. For example, when referencing previous studies on the reliability of interplanetary missions, it would be useful to briefly summarize the findings of those studies. Below are some suggestions for addition.

https://doi.org/10.1016/j.ssci.2013.04.011

doi: 10.1109/IBCAST.2014.6778141  

·         The section could benefit from a clearer organization of the different topics discussed. While the section covers many important points, it jumps from one topic to another without a clear structure or hierarchy.

·         It would be useful to provide more information on the current state-of-the-art in terms of reliability assessment and life prediction methods for extravehicular spacesuits. This would help set the stage for the research questions addressed later in the paper.

·         Which software has been used for the calculation of failure rate?

·         Revise conclusion by adding some findings.

Revision required

Reviewer 2 Report

1.     There should be some introduction about materials from which the particular parts of the suit are built. Especially in the part where the authors are explaining the influence of fine movement and friction on joints (Section 2.1.). There is a mention of soft structures and their tendency to degradation based on movements, but there is no explanation of soft structures. From what kind of materials are they built? The same issue is in the part about temerature influence and termal degradation, there is no mention about type of material that is being influenced. Only mention of materials in general context of rubber materials is in the section 3.1. where the Arrhenius formula in introduced, which is very little and insufficient considering the topis of the research.

2.      In the section 3.4. authors are quoting some assumptions for parameters like design life, number of wears and joint movements that are integrated in their the simulation program. Based on what are those assumptions made? Literature, previous research, experiental knowledge... If so, there should be a citation or at least a mention how they came to these assumtions.

3.       In general, this is a very interesting research with great potential of useful application and it would be of interest to journal readers and other researchers from the aeronautic field.

Round 2

Reviewer 1 Report

Accepted: Revised version 

Reviewer 2 Report

The text has been revised by authors. I am satisfied with the authors response to my comments and recommend to accept the manuscript for publication.